# Using Porcine Jejunum Ex Vivo to Study Absorption and Biotransformation of Natural Products in Plant Extracts: *Pueraria lobata* as a Case Study

**DOI:** 10.3390/metabo11080541

**Published:** 2021-08-14

**Authors:** Joëlle Houriet, Yvonne E. Arnold, Léonie Pellissier, Yogeshvar N. Kalia, Jean-Luc Wolfender

**Affiliations:** 1School of Pharmaceutical Sciences, University of Geneva, Centre Médical Universitaire, Rue Michel-Servet 1, 1211 Geneva, Switzerland; joelle.houriet@unige.ch (J.H.); y.arnold@gmx.ch (Y.E.A.); Leonie.Pellissier@unige.ch (L.P.); yogi.kalia@unige.ch (Y.N.K.); 2Institute of Pharmaceutical Sciences of Western Switzerland, University of Geneva, Centre Médical Universitaire, Rue Michel-Servet 1, 1211 Geneva, Switzerland

**Keywords:** isoflavones, *Pueraria lobata*, intestinal permeability, intestinal biotransformation, Ussing chamber, untargeted mass spectrometry, feature-based molecular network

## Abstract

Herbal preparations (HPs) used in folk medicine are complex mixtures of natural products (NPs). Their efficacy in vivo after ingestion depends on the uptake of the active ingredient, and, in some cases, their metabolites, in the gastrointestinal tract. Thus, correlating bioactivities measured in vitro and efficacy in vivo is a challenge. An extract of *Pueraria lobata* rich in different types of isoflavones was used to evaluate the capacity of viable porcine small intestine ex vivo to elucidate the absorption of HP constituents, and, in some cases, their metabolites. The identification and transport of permeants across the jejunum was monitored by liquid chromatography-mass spectrometry (LC-MS), combining targeted and untargeted metabolite profiling approaches. It was observed that the C-glycoside isoflavones were stable and crossed the intestinal membrane, while various O-glycoside isoflavones were metabolized into their corresponding aglycones, which were then absorbed. These results are consistent with human data, highlighting the potential of using this approach. A thorough investigation of the impact of absorption and biotransformation was obtained without in vivo studies. The combination of qualitative untargeted and quantitative targeted LC-MS methods effectively monitored a large number of NPs and their metabolites, which is essential for research on HPs.

## 1. Introduction

Plant extracts are often used by the population, both in traditional medicines and through the food supplement market. Standardized extracts with clinical proof of efficacy are also registered by authorities as drugs in the form of herbal drug preparations. The medical uses of these herbal drug preparations are based on modern evidence-based approaches [1,2]. Medicinal plants contain chemo-diverse molecules (natural products (NPs)) that may exhibit strong bioactivities, and some are used as drugs [3]. Although the World Health Organization recommends evaluating traditional and popular practices as potential sources of new treatments based on plant extracts [4], little research is being conducted to understand the mechanism of action of these complex preparations at the molecular level, which is, however, desirable in order to justify their use with scientific criteria [5].

In pharmacognosy and ethnopharmacology, many plant extracts, as well as pure NPs, have been evaluated for biological activities with bioassays in vitro. Those tests are relevant from a drug-discovery perspective to identify a pure product (mono substance) and highlight compounds that, after formulation, may become drugs. In studies aimed at understanding the clinical efficacy of herbal preparations (HPs), notably taken in the form of infusions or decoctions, the link between the in vitro activity of individual plant constituents and pharmacological activity of the preparations in vivo is often unclear or not known [6,7]. Indeed, such preparations are ingested, implying that their constituents will be subjected to absorption phenomena and metabolism. Studying these pharmacokinetic (PK) processes is a critical step in deciphering the mechanism of action of these HPs [6,7]. In drug discovery for single substances, PK studies are an integral part of the process. In contrast, in pharmacognosy, few PK studies have involved NPs and, even fewer have investigated extracts. However, PK studies are required to determine which NPs are capable of reaching the systemic circulation to focus the in vitro pharmacological evaluation on the absorbed NPs and, thereby, identify the mode of action.

As constituents of foods or HPs, NPs enter the human body mainly through ingestion; thus, absorption and intestinal biotransformation are critical steps in determining any pharmacological effect of the ingested substances. Regarding absorption, the complex structure of the intestinal wall contains several distinct absorption pathways, including passive transport, which can be trans- and/or paracellular, and active transport via carrier-mediated processes [8]. For biotransformation, the intestine is considered the second most important site after the liver and contains phase 1 and 2 enzymes, including the cytochrome P450s (CYPs) [9,10].

Considering both the complexity of absorption pathways and plant-based extracts’ chemical composition, studying absorption is challenging. Today, few models can explore the bioavailability of NPs ingested as complex mixtures. In the field of research dedicated to traditional Chinese medicine (TCM), numerous in vivo PK studies have recently been performed and have provided a large amount of valuable information [11]. However, in vivo studies on laboratory animals are challenging to implement, notably for ethical reasons. Furthermore, animal models differ significantly from the human body, making extrapolations sometimes difficult [9]. Therefore, in view of reducing the number of in vivo studies, summarized by the 3R rules (Replacement, Reduction, and Refinement) [12], developing efficient in vitro/ex vivo models for a preliminary evaluation of ADME is necessary.

Several in vitro models were used so far to study the absorption of NPs. Among them, the simplest is the Parallel Artificial Membrane Permeability Assay (PAMPA). It evaluates the passive absorption of NPs alone or included in HPs and has the advantage of being a high-throughput assay [13]. The Caco-2 cell line model is a more complex model than PAMPA but has the advantage of evidencing active transport mechanisms [14]. It has been used to evaluate the absorption of NPs alone [15,16] or from their source plants, including mixtures of plants [17,18,19]. The main limitations of Caco-2 cell line are the absence of the different characteristics of each intestinal segment (duodenum, jejunum, and ileum), the low level of the main enzyme of phase 1 metabolism (P450 CYP3A4), and the overexpression of the efflux transporter P-glycoprotein (P-gp) [20]. Another approach is to use ex vivo intestinal tissues inserted into an Ussing chamber system, allowing the continuous monitoring of the transepithelial electrical resistance (TEER) as an integrity indicator of the intestinal tissue [20,21]. For human drug discovery studies, the intestines of various species have been used in Ussing chamber system, including human [20,22,23,24], rat [25,26,27], dog [27,28], monkey [28], and pig [23,29]. Since human intestine is difficult to access, several studies have compared the intestines of these different species, observing that pig intestine has a greater physiological similarity to human intestine [23,30]. Both rat [31,32,33,34] and porcine tissue [35,36,37] have been used in studies involving pure NPs and plant extracts.

Recently, porcine intestine mounted in an Ussing chamber system was used to measure intestinal drug permeability coefficients (*P_app, pig_*) for a series of drugs for which in vivo human permeability coefficients (*P_eff, human_*) had been previously described [29]. The work highlighted the robustness of this model to study absorption, particularly by demonstrating the high correlation between *P_app, pig_* and *P_eff, human_*, and showed that P450 CYP3A4 and P-gp activity was present. The presence and activity of membrane transport proteins was investigated in more depth in a subsequent study [38]. The presence of presystemic metabolic activity and the presence of functional membrane transporters is essential for reliable prediction of the uptake of NPs from a HP in the human intestine since NPs can serve as prodrugs, which are biotransformed by enzymes present in the intestinal wall [7].

In this study, we aimed to evaluate whether porcine intestine ex vivo could be used to investigate the absorption of NPs contained in a traditional HP. For this, we used (i) untargeted metabolomics for a qualitative monitoring of the molecules prior, during, and after intestinal membrane passage and (ii) targeted quantitative measurement for determining the permeability coefficients of selected molecules. For the untargeted detection of the largest possible number of NPs, advanced untargeted metabolite profiling methods with high sensitivity and high resolution were used. Such methods, described in a detailed review by Wolfender et al. [39], enable a comprehensive coverage of NPs, and identification is performed through comparison with in silico spectral libraries and can be organized and visualized by molecular networking [40,41,42]. To our knowledge, all previous studies aiming at predicting NPs permeability with the Ussing chamber system used targeted analytical methods.

The root of *Pueraria montana* var. *lobata* (Willd.) Sanjappa & Pradeep (PL) was selected for this study since this plant and/or its constituents have been investigated in previous in vivo metabolism and absorption studies. PL is an important herb in TCM and has a well-characterized composition with only a few main phenolic constituents [43]. It is known to contain various types of isoflavones, such as *O*-glycosides, *C*-glycosides, and related aglycones, reported to be absorbed via different pathways [44,45,46]. These isoflavones are not restricted to PL species and are found in various herbs, notably in the widely studied soya (*Glycine max* (L.) Merr.) [47] and the clover (*Trifolium pretense* L.) [48]. As an important dietary food in many Asian countries, soya has been the subject of numerous studies, including clinical studies, which have described the human pharmacokinetic data of its main constituents [49,50,51,52]. Furthermore, a comparative study of isoflavone absorption on different animal tissues was performed, highlighting the similarities between human and porcine PKs and the differences with rats and mice [53]. Taken together, these studies supported the selection of PL for testing with the porcine intestine in the Ussing chamber system.

In this study, we used the freeze-dried decoction of the root of PL to evaluate the feasibility of using porcine small intestine ex vivo in the Ussing chamber system for investigating the intestinal absorption and biotransformation of the largest possible NPs present in the extract. A combination of untargeted and targeted MS-based metabolomics approaches was used to identify NPs and/or their metabolites deposited in or permeated across the intestine. This method could provide an overview of the NPs that are capable of reaching the bloodstream, either as such, or as products of intestinal biotransformation.

## 2. Results

### 2.1. General Overview of Ussing Chamber System

The Ussing chamber system was used to study the absorption and intestinal biotransformation of the constituents of PL root extract (PLRE). This device consists of donor and acceptor compartments, separated by an insert in which the biological tissue is mounted (Appendix A). The membrane used in this study came from the jejunum of adult pigs. The jejunum was collected in a slaughterhouse immediately after slaughter, placed in a physiological buffer (Krebs-Bicarbonate Ringer buffer (KBR)), and oxygenated during transport to the laboratory in order to maintain its viability. To study the permeation of given constituents and/or complex plant extract, the selected sample was solubilized in physiological buffer and added to the donor compartment of the Ussing chambers. The acceptor compartment also filled with the buffer was separated from the donor by the intestinal membrane. Both compartments were oxygenated until completion of the permeation experiment to ensure membrane viability and buffer circulation. These general conditions corresponded to those employed for the permeation assessment of common drugs [29].

### 2.2. Metabolite Profiling of the Pueraria lobata Root Extract

Before the permeation experiments, the roots of PL were extracted by decoction according to the traditional protocol and then freeze-dried. This PLRE mimicked what is usually ingested orally when drinking the decoction [43]. The composition of this extract was assessed with a generic untargeted reversed-phase qualitative metabolite profiling method with different detections (for details on data processing, see Section 4.8.1). On the one hand, UHPLC-HRMS/MS-UV-PDA was used for the identification of its constituents (Figure 1C,D, and Appendix A), and, on the other, UHPLC-UV-PDA coupled to an evaporative light scattering detector (ELSD) was employed for a semi-quantitative estimation of their proportion (Figure 1A,B). To annotate all peaks detected in UHPLC-HRMS/MS profiles, both fragmentation HRMS/MS and HRMS spectra were considered and compared with previously reported PL constituents (Appendix A). Thus, the annotation was based on molecular formula assignment and comparison of all HRMS/MS spectra against a database of in silico fragmented spectra for all known plant secondary metabolites [40,54]. The annotation results were weighted based on taxonomy (previous occurrence in *Pueraria lobata*, in its genus and family) [55], and the HRMS/MS spectra were organized as a feature-based molecular network (FBMN) for verifying the consistency of the annotations. The resulting FBMN contained 797 nodes, organized in 40 clusters (Appendix A). Since constituents may generate different type of molecular ion species (e.g., adducts and dimers), the number of nodes does not reflect the numbers of constituents.

**Figure 1 metabolites-11-00541-f001:**
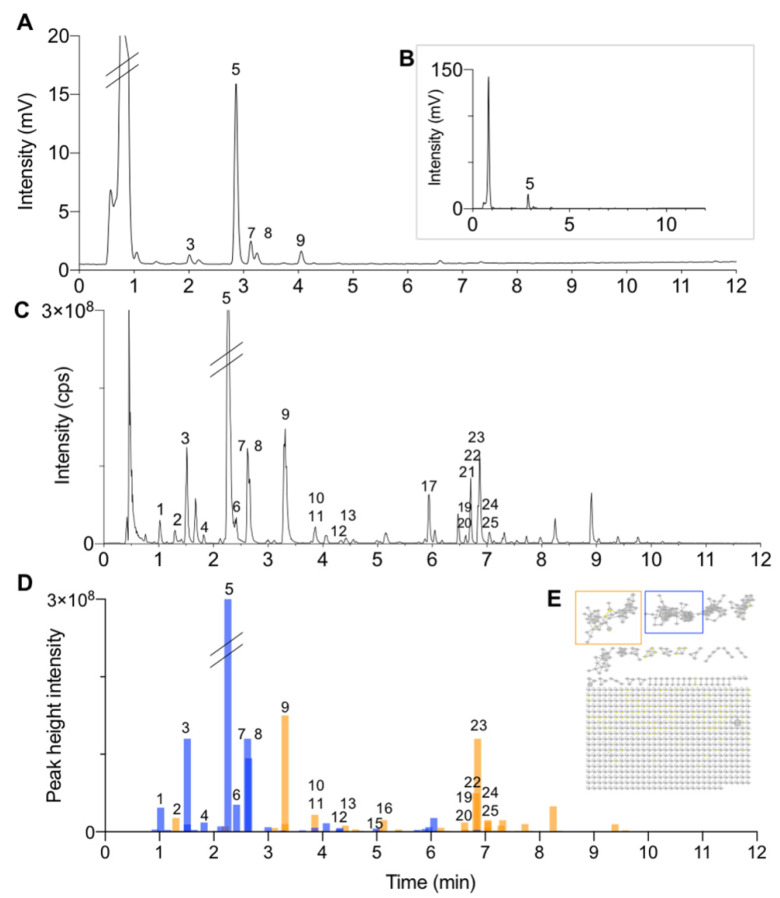
Metabolite profiling of PLRE: (**A**,**B**) UHPLC-ELSD chromatogram highlighting the main constituents of PLRE, (**C**) UHPLC-HRMS/MS chromatogram in ESI positive ionization, and (**D**) bar chromatogram of the peaks detected in two distinct clusters in the FBMN (**E**). (**E**) FBMN of the metabolite profiling of PLRE. In (**D**,**E**), in blue, the peaks of the cluster containing the *C*-glycoside isoflavone puerarin (**5**) and linked peaks; in orange, the peaks of the cluster containing isoflavone *O*-glycosides and aglycones, including daidzin (**9**), genistin (**16**), daidzein (**23**), and genistein (**28**). See Table 1, Table 2, Table 3 and Table 4 for annotations and Appendix A for the full FBMN.

In parallel to the UHPLC-HRMS/MS profiling, the detection by ELSD was used to highlight the main PLRE constituents. This detection is less sensitive and known to provide a rather equivalent factor-response for semi-quantitative evaluation [56]. The ELSD profile revealed the high proportion of polar compounds not retained in reversed-phase (mainly sugars), as well as five main ELSD peaks corresponding to secondary metabolites (NPs of interest) (**3**, **5**, **7–9**) (Figure 1A,B).

**Table 1 metabolites-11-00541-t001:** Annotation of PLRE constituents and its intestinal metabolites.

N°	*m*/*z* [M + H]^+^	Retention Time (min)		Molecular Formula	PPM ^a^	Annotation	Taxonomy	MSI ^b^	LogP ^c^
		Figure 1	Figure 2						
**1**	579.1711	1.02	0.86	C_27_H_30_O_14_	0.4	puerarin-6″-*O*-glucoside	Species	2	
**2**	579.1711	1.30	1.10	C_27_H_30_O_14_	0.4	daidzin-4″-glucoside	Species	2	
**3**	433.1131	1.51	1.31	C_21_H_20_O_10_	0.4	3′-hydroxypuerarin	Species	2	
**4**	565.1554	1.82	1.52	C_26_H_28_O_14_	0.5	genistein-8-*C*-apioside	Species	2	
**5**	417.1181	2.20	1.97	C_21_H_20_O_9_	0.2	puerarin	Species	1	−0.03
**6**	549.1605	2.42	2.05	C_26_H_28_O_13_	0.4	mirificin (puerarin apioside)	Species	2	
**7**	549.1605	2.64	2.27	C_26_H_28_O_13_	0.4	6″-xylo-puerarin	Species	2	
**8**	447.1287	2.62	2.30	C_22_H_22_O_10_	0.2	3′-methoxypuerarin	Species	2	
**9**	417.1181	3.31	2.93	C_21_H_20_O_9_	0.2	daidzin	Species	1	0.46
**10**	447.1287	3.89	3.45	C_22_H_22_O_10_	0.2	3′-methoxydaidzin	Species	2	
**11**	433.1131	3.86	3.45	C_21_H_20_O_10_	0.4	genistein-8-*C*-glucoside	Species	2	
**12**	417.1181	4.32	3.88	C_21_H_20_O_9_	0.2	neopuerarin A	Species	2	
**13**	417.1181	4.42	3.99	C_21_H_20_O_9_	0.2	daidzein-4′-glucoside	Family	2	
**14**	607.2024	4.56	4.10	C_29_H_34_O_14_	0.4	pueroside A	Species	2	
**15**	417.1181	4.99	4.52	C_21_H_20_O_9_	0.2	neopuerarin B	Species	2	
**16**	433.1131	5.13	4.71	C_21_H_20_O_10_	0.4	genistin	Family	1	0.81
**17**	503.1186	5.94	5.53	C_24_H_22_O_12_	0.3	6″-*O*-malonyldaidzin	Species	2	
**18**	271.0601	6.14	5.85	C_15_H_10_O_5_	−0.1	8-hydroxydaidzein	Species	2	2.43
**19**	459.1287	6.71	6.33	C_23_H_22_O_10_	0.2	6″-*O*-acetyldaidzin	Family	2	
**20**	519.1135	6.62	6.48	C_24_H_22_O_13_	0.3	6″-*O*-malonylgenistin	Family	2	
**21**	475.1602	6.71	6.58	C_24_H_26_O_10_	0.4	pueroside D	Species	2	
**22**	431.1339	6.83	6.72	C_22_H_22_O_9_	0.5	-formononetin-7-*O*-galactoside -ononin (7-*O*-glucoside)	Family Species	3	
**23**	255.0652	6.86	6.74	C_15_H_10_O_4_	0.0	daidzein	Family	1	2.73
**24**	285.0758	7.05	6.92	C_16_H_12_O_5_	0.1	3′-methoxydaidzein	Family	2	2.57
**25**	431.1339	7.05	6.96	C_22_H_22_O_9_	0.5	-formononetin-7-*O*-galactoside -ononin (7-*O*-glucoside)	Family Species	3	
**26**	285.0758	7.19	7.08	C_16_H_12_O_5_	0.1	kakkatin	Genus	2	2.57
**27**	447.1288	7.67	7.46	C_22_H_22_O_10_	0.5	Sissotrin (biochanin A-7-*O*-glucoside)	Genus	2	
**28**	271.0601	7.60	7.50	C_15_H_10_O_5_	−0.1	genistein	Family	1	3.08
**29**	313.107	7.75	7.66	C_18_H_16_O_5_	−0.2	puerol B	Species	2	
**30**	269.0809	8.25	8.16	C_16_H_12_O_4_	0.2	formononetin	Family	2	2.98
**31**	299.0915	8.36	8.20	C_17_H_14_O_5_	0.3	tithonin	Family	2	
**32**	285.0759	ND	8.96	C_16_H_12_O_5_	0.4	biochanin A	Family	2	3.22

^a^ calculated with the prediction module of MZmine 2 for the dataset used for Figure 2, ^b^ confidence level in accordance with the Metabolomics Standards Initiative (MSI) [57], ^c^ consensus LogP calculated with Reference [58].

**Table 2 metabolites-11-00541-t002:** Annotated isoflavone aglycones.

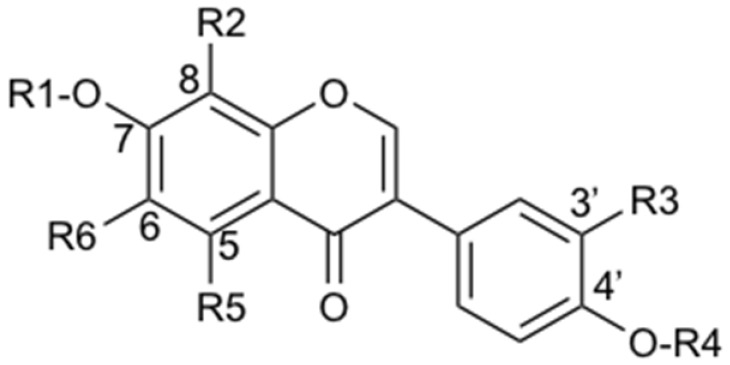							
N°	Compound Name	R1	R2	R3	R4	R5	R6
**23**	**daidzein**	**H**	**H**	**H**	**H**	**H**	**H**
**18**	8-hydroxydaidzein	H	OH	H	H	H	H
**24**	3′-methoxydaidzein	H	H	O–CH_3_	H	H	
**26**	kakkatin	CH_3_	H	H	H	H	OH
**28**	**genistein**	**H**	**H**	**H**	**H**	**OH**	**H**
**30**	formononetin	H	H	H	CH_3_	H	H
**31**	tithonin	CH_3_	H	H	CH_3_	H	H
**32**	biochanin A	H	H	H	CH_3_	OH	H

In bold: the identity has been verified by comparison with a commercial standard.

**Table 3 metabolites-11-00541-t003:** Structures of annotated isoflavone *C*-glycosides.

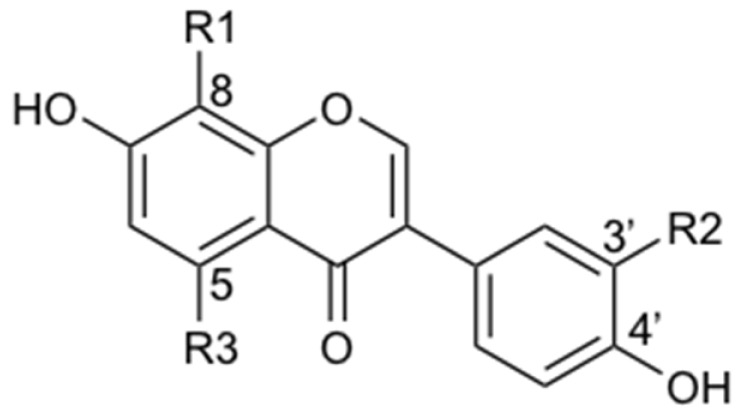				
N°	Compound Name	R1	R2	R3
**5**	**puerarin**	***C*-glucosyl**	**H**	**H**
**1**	puerarin-6″-*O*-glucoside	*C*-glucosyl-6″-*O*-glucosyl	H	H
**3**	3′-hydroxypuerarin	*C*-glucosyl	OH	H
**4**	genistein-8-*C*-apioside	*C*-glucosyl-6″-*O*-apiosyl	H	OH
**6**	mirificin	*C*-glucosyl-6″-*O*-apiosyl	H	H
**7**	6″-xylo-puerarin	*C*-glucosyl-6″-*O*-xylosyl	H	H
**8**	3′-methoxypuerarin	*C*-glucosyl	O-CH_3_	H
**11**	genistein-8-*C*-glucoside	*C*-glucosyl	H	OH
**12**	neopuerarin A	*C*-α-glucofuranosyl	H	H
**15**	neopuerarin B	*C*-β-glucofuranosyl	H	H

In bold: the identity has been verified by comparison with a commercial standard.

**Table 4 metabolites-11-00541-t004:** Structure of annotated isoflavone *O*-glycosides.

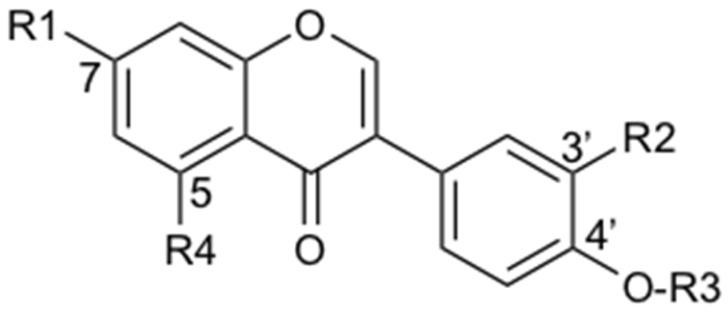						
N°	Compound Name	R1	R2	R3	R4	R5
**9**	**daidzin**	***O*-glucosyl**	**H**	**H**	**H**	**H**
**2**	daidzin-4″-glucoside	*O*-glucosyl-4″-*O*-glucosyl	H	H	H	H
**10**	3′-methoxydaidzin	*O*-glucosyl	O-CH_3_	H	H	H
**13**	daidzein-4′-glucoside	OH	H	O-glycosyl	H	H
**16**	**genistin**	***O*-glucosyl**	**H**	**H**	**OH**	**H**
**17**	6″-*O*-malonyldaidzin	*O*-glucosyl-6″-*O*-malonyl	H	H	H	H
**19**	6″-*O*-acetyldaidzin	*O*-glucosyl-6″-*O*-acetyl	H	H	H	H
**20**	6″-*O*-malonylgenistin	*O*-glucosyl-6″-*O*-malonyl	H	H	OH	H
**22–25**	Ononin formononetin-7-*O*-galactoside	*O*-glucosyl *O*-galactosyl	H	O-CH_3_	H	H
**26**	kakkatin					
**27**	sissotrin	*O*-glucosyl	H	O-CH_3_	OH	

In bold: the identity has been verified by comparison with a commercial standard.

Since PL is a widely used medicinal herb whose composition has been well documented [43], an annotation strategy was performed first. The main NP was annotated as puerarin (**5**), a predominant *C*-glucoside isoflavone described for PL [43] (Table 1 and Table 3 and Figure 1). Another main constituent was annotated as daidzin (**9**), an O-glucoside sharing the same aglycone as puerarin (daidzein (**23**)) (Table 2 and Table 4). Nine additional *C*-glycosides (**1**, **3**, **4**, **6**, **7**, **8**, **11**, **12**, **15**) and nine *O*-glycosides (**2**, **10**, **13**, **16**, **17**, **19**, **20**, **22**, **25**) were annotated from these two classes of glycosides, of which **3**, **7**, **8** were clearly detected by ELSD (Figure 1C,D and Table 1, Table 3 and Table 4). Thus, the five secondary metabolites detected by ELSD were all isoflavone glycosides, which is logical since the extraction was obtained by decoction. Two puerosides were additionally annotated (**14**, **21**) (Table 1). Finally, the corresponding aglycones, mainly daidzein (**23**), were detected (Table 1 and Table 2). Among these annotations, five were compared with commercial standards, allowing their unambiguous identification (puerarin (**5**), daidzin (**9**), genistin (**16**), daidzein (**23**), and genistein (**28**)).

### 2.3. Establishing UHPLC-HRMS Conditions for Profiling of PLRE during the Permeation Experiment

The permeation experiment had to be adapted for a direct investigation of an HP crude extract and the untargeted detection of its multiple constituents. For this, the concentration and solubility of PLRE constituents in the chosen buffer (KBR) had to be carefully considered. On the one hand, the concentration of PLRE constituents should not be too high to avoid saturating the intestinal membrane, and on the other hand, it should be high enough to allow the detection of compounds at low concentrations.

For PLRE, the selection of the extract concentration was based on the level of its main constituents (puerarin (**5**) and daidzin (**9**)). To that end, these constituents were quantified in PLRE by targeted UHPLC-MS using a triple quadrupole operating in Multiple Reaction Monitoring (TQ-MRM) (Section 4.9). It has to be noted that compounds showing a concentration ratio between the acceptor compartment (100 min) and donor compartment (0 min) of about 0.1% are still considered as being permeants [59]. Having this limit in mind, the limit of detection (LOD) by untargeted UHPLC-HRMS method were assessed for puerarin and daidzin and found to be in the range of 2 to 8 nM (Section 4.8.2). Consequently, their initial levels in PLRE donor compartment had to be ideally above 2 µM. Taking in consideration all this preliminary data, an initial concentration of PLRE in the buffer was set at 200 µg/mL (see the corresponding metabolite profile in Figure 2A). No solubility issue was encountered at this concentration, as PLRE is a water extract (decoction) composed largely of sugars (Figure 1A,B). At 200 µg/mL, the levels of puerarin and daidzin were above 2 µM, at 25.724 ± 0.288 µM and 5.035 ± 0.040 µM, respectively (Section 4.8 and Appendix A). Such a range of concentration is slightly lower than the one used for validating the intestinal ex vivo model with known drugs [29]. Based on the determined LOD, such concentrations were considered compatible for detection with the method used in this study.

The evaluation of other constituents’ concentration was obtained by comparing the ELSD response of these standards in the PLRE ELSD profile (Figure 1B), which revealed that other isoflavones (**3**, **7**, **8**) were also in a similar detectable range. This was sufficient for a first estimation of the permeation behavior of all main constituents of the extract.

### 2.4. Measurement of Permeation of the Constituents of PLRE

The PLRE solubilized in KBR buffer was exposed to the living intestinal membrane in the Ussing chambers at the selected concentration (200 µg/mL) for 100 min to allow the interaction of the constituents with the membrane. Although it has previously been shown that intestinal tissue viability is maintained for 120 min, given the complex nature of the PLRE, and an eventual increased risk of degradation, it was decided to use the shorter application time of 100 min [29].

After completing the experiment, the contents of the donor and acceptor compartments were collected. In addition, the intestinal membranes were extracted to determine the amounts retained within the membrane. Compounds with a detectable concentration in the membrane are considered to be able to permeate in vivo—the limited duration of the experiment that is required to ensure that the viability of the membrane is maintained throughout the experiment, means that these compounds do not have sufficient time to diffuse across the membrane and into the acceptor compartment [24,27,29].

Four replicates of the experiment were performed, and both compartments and membranes, as well as the initial stock solution, were profiled by UHPLC-HRMS, together with relevant blank solutions (see Section 4.8.2).

### 2.5. Evaluation and Comparison of the Different Untargeted Metabolite Profiles

#### 2.5.1. PLRE Constituents in the Acceptor Compartment and in the Intestinal Membrane

First, only the *C*-glucoside puerarin (**5**) was detected in the acceptor compartment, showing its effective permeation through the intestinal membrane (Appendix A). Next, the content of the extracted membrane was analyzed to identify constituents that penetrated it. This profile was compared to a blank extract of the membrane to avoid any interference with micronutrients found in the pig diet. In addition to puerarin, two other *C*-glycoside derivatives (6″-xylopuerarin (**7**) and 3′-methoxypuerarin (**8**)) were detected, as well as two aglycones isoflavones (daidzein (**23**) and genistein (**28**)) (Figure 1, Table 1, Table 2 and Table 3, Appendix A). A semi-quantitative evaluation was provided by calculating the ratio between the peak heights detected in the membrane and the initial stock solution expressed as percentage (Section 4.8.2, Equation (3)). For the *C*-glycosides isoflavones (**5, 7, 8**), these ratios indicated that between 0.3% and 0.6% of the initial amount in the stock solution entered the jejunum. Interestingly, these ratios were significantly higher for both aglycones. For **23**, 13.4% of the initial amount was present in the membrane (Figure 2D and Appendix A). For **28**, surprisingly, this ratio reached 116% ratio. Since we could demonstrate that no significant interference with possible dietary isoflavones occurred, this clearly indicated that this high amount had to come from possible biotransformation of PLRE constituents.

#### 2.5.2. Fate of PLRE Constituents in the Donor Compartments

To further interpret the membrane data, the UHPLC-HRMS profile of the initial stock solution was systematically compared with the donor compartment after 100 min (Figure 2B,C). A first visual inspection revealed that peaks corresponding to *C*-glycoside isoflavones (**5**, **7**, **8**) were mostly unchanged between the initial stock solution and the donor compartment at 100 min. In contrast, the peak corresponding to the main *O*-glucoside daidzin (**9**) disappeared, and an intense peak corresponding to its aglycone daidzein (**23**) appeared. These observations revealed that the constituents of PLRE had undergone biotransformation once in the presence of jejunum. The ratios between the peak heights in the donor compartments after 100 min of membrane exposure and the initial stock solution were calculated (Section 4.8.2, Equation (5) and Appendix A). For the two aglycones detected in the membrane, this semi-quantitative estimation showed that the amount of daidzein (**23**) increased by about seven times in the donor compartment (774%), and the one of genistein (**28**) increased by more than 30 times (3736%).

To understand all constituents’ fate in the donor compartment, the various corresponding features of UHPLC-HRMS/MS metabolite profiles were peak-picked and aligned with those of the initial stock solution to generate a unique peak list. As previously described for the main constituents (Section 2.2), the HRMS/MS data corresponding to these peaks were used to annotate the constituents found in the stock solution and in the donor compartments after 100 min. This second annotation process enabled the annotation of eight additional NPs, and a total of 32 NPs were annotated before and after exposure to the membrane (for details on data processing, see Section 4.8.2, as well as Table 1, Table 2, Table 3 and Table 4 for the annotations). A detailed comparison of all these constituents revealed an increase in several NPs, which were, therefore, considered as intestinal metabolites. The results were consistent across the replicates and are detailed by types of glycosides present in PLRE in the following sections. 

#### 2.5.3. Fate of the *C*-glycosides in the Donor Compartment

As it was the case for puerarin (**5**), a detailed analysis of all *C*-glycoside isoflavones (**1**, **3**, **4, 6**, **7**, **8**, **11**, **12**, **15**) revealed that they were mainly stable once in contact with the small intestine in the donor compartment (**1**, **6**, **7**, **8**, **11**, **12**, **15**) (Figure 2B,C, Table 1 and Table 3, and Appendix A). Only a C-glycoside disaccharide derivative (**1**) was found less stable with a ratio of 85.9% (Section 4.8.2, Equation (5)). On the other hand, surprisingly, two *C*-glycosides (3′-hydroxypuerarin (**3**) and genistein-8-*C*-apioglucoside (**4**)) were observed with a ratio of about 200%. These two compounds have three hydroxy groups on their aglycone. Thus, their increase in the presence of the membrane was most likely due to an enzymatic hydroxylation, probably by the cytochrome CYP3A4 still active in this setting [29]. Such hydroxylation could occur on various dihydroxy-*C*-glycoside isoflavones with a free 3′ or 5 position (**1**, **5**, **6**, **7**, **12**, **15**) (Table 1 and Table 3).

#### 2.5.4. Fate of the *O*-glycosides in the Donor Compartment

Concerning *O*-glucosides isoflavones, their levels dropped drastically in the donor compartment once in the presence of the small intestine. This biotransformation concerned the main *O*-glucoside daidzin (**9**), as well as (**2**, **10**, **13**, **16**, **19**, **22**, **25**) (Figure 2B,C, Table 1 and Table 4, and Appendix A). In parallel, the corresponding aglycone levels increased (Table 2 and Appendix A). In addition to daidzein (**23**) and genistein (**28**), the following aglycones were detected: formononetin (**30**), 3′-methoxydaidzein (**24**), biochanin A (**32**), kakkatin (**26**), and tithonin (**31**). Thus, all these aglycones resulted from intestinal biotransformation as they were either not detected or present only in trace amounts in PLRE. These observations demonstrated that the high level of daidzein in the membrane could be related to hydrolysis of the mono-glycosides (**9**, **13**), di-glucoside (**2**), and acetylglucoside (**19**) present in PLRE.

A notable exception is the case of malonyl-glycosides isoflavones (**17, 20**) (Table 4). These constituents were much more stable than all other *O*-glycosides and were still present at a ratio of 84% (Section 4.8.2, Equation (5)).

Interestingly, one aglycone, 3′-hydroxydaidzein (**18**), also appeared in the donor compartment (Table 4). Since no corresponding *O*-glycoside precursors were detected, this can be interpreted as a sign of possible CYP3A4 enzymatic activity on daidzein, similar to what was observed for the tri-hydroxylated *C*-glycosides (**3**, **4**).

#### 2.5.5. Isoflavones Detected in the Intestinal Membrane and in the Acceptor Compartment

Based on this detailed analysis on the fate of all isoflavone derivatives in the donor compartment, the presence of both intact and biotransformed metabolites was systematically verified in the extracted membrane and the acceptor compartment (Figure 2D, Table 1, Table 2, Table 3 and Table 4, and Appendix A). To that end, since sensitive detection is required, all compounds were detected by “Targeted peak deconvolution” (Section 4.8.2). The aligned peak list obtained on all samples was used to systematically compare the intensity of the features corresponding to the targeted metabolites. Given the observations made on the donor compartment, the ratio between the peaks detected in the membrane and the donor compartment at 100 min was calculated to consider aglycones produced by biotransformation (Section 4.8.2, Equation (4), and Appendix A). This analysis revealed that, in the extracted membrane, the aglycones (at least partially resulting from biotransformation) formononetin (**30**), 3′-methoxydaidzein (**24**), tithonin (**31**), and biochanin A (**32**) were present with ratios of 3.0%, 0.7%, 3.7%, and 9.0% respectively, in addition to daidzein (**23**) and genistein (**28**), which had ratios of 1.7% and 3.1%, respectively (Table 1 and Table 2). The stable *C*-glycosides 3′-methoxypuerarin (**8**) and 6″-xylopuerarin (**7**) were confirmed with a ratio of 0.6% and 0.3%, in addition to puerarin, which had a ratio of 0.5% (Table 3).

The same analysis for the acceptor compartment revealed no compounds other than puerarin (**5**). Trace amounts of daidzein (**23**) and 3′-methoxypuerarin (**8**) at the limit of detection were observed.

Thus, a total of nine NPs (three *C*-glycoside isoflavones (**5**, **7**, **8**) and five aglycones were observed in the extracted membranes (**23**, **24**, **30**, **31, 32**) and only one in the acceptor compartment (**5**). These observations highlighted the limited number of PLRE constituents likely to reach plasma circulation in the jejunum. They also demonstrated that biotransformation processes have an essential role in the case of *O*-glycosylated isoflavones. In addition, these observations demonstrated the viability of the intestinal membrane, not only in terms of transporters but also of biotransformation enzymes.

### 2.6. Quantitative Measurements

All results discussed so far were only based on semi-quantitative ratios and did not allow measurement of the permeability coefficient (*P_app, pig_*) and the Transport Index (TI) ratios (Section 4.10). The permeability coefficient *P_app, pig_* reflects the rate at which a given drug crosses the membrane [14] (Section 4.10.1, Equation (6)). The concept of the Transport index (TI) was recently proposed to consider the retention of compounds in the intestinal membrane [24]. *TI* is defined as the sum of the quantity of substance transported in the acceptor compartment (permeating quantity *Q**_PERM_* (Section 4.10.2, Equation (8))) and accumulated in the intestinal tissue (quantity deposited *Q**_DEP_* (Section 4.10.2, Equation (7))). Targeted quantitative measurements were, therefore, performed in a second step.

In this respect, the untargeted profiles demonstrated that the behavior of puerarin (**5**), daidzein (**23**), and genistein (**28**) could be evaluated in more detail. Puerarin (**5**) was detected in the acceptor compartment, while daidzein (**23**) and genistein (**28**) resulting from intestinal biotransformation were observed in the donor compartment and the intestinal membrane. These three NPs, as well as the corresponding *O*-glucosides (daidzin (**9**) for daidzein (**23**), and genistin (**16**) for genistein (**28**)) were, therefore, quantified by external calibration with targeted UHPLC-TQ-MRM (Section 4.9).

In addition, the fate of two pure standards was studied in the same experimental setting to verify the consistency of the *P_app, pig_* and TI measured in PLRE. The *C*-glucoside puerarin (**5**) was selected as a permeant constituent, and the *O*-glucoside daidzin (**9**) as a constituent that undergoes biotransformation (daidzin (**9**) -> daidzein (**23**)). Given the semi-quantitative results, the classical parameter, i.e., the permeability coefficient (*P_app, pig_*) (Equation (6)) was measured for puerarin (**5**). The ratios used in the TI approach were also calculated to consider the amount retained in the intestinal membrane (*Q_DEP_*) (Equation (7)) in addition to the amount which permeated across the membrane (*Q_PERM_*) (Equation (8)). This last approach was also used for the pair daidzin (**8**) -> daidzein (**23**), as no daidzein was detected in the acceptor compartment. The Metabolites Formation Index (MFI) was eventually calculated to evaluate daidzein’s biotransformation from the pure daidzin (**8**) (Section 4.10.3, Equation (10)) [59]. If these calculations are suitable for assessing a mono substance’s fate, they had to be adapted to an extract containing many constituents. Indeed, it has been observed that, in addition to daidzin (**9**), different constituents (**2**, **13**, **19**) could be biotransformed into daidzein (**23**). In this context, we proposed to evaluate the amount of daidzein (**23**) penetrating the membrane independently of the initial amounts of the different constituents potentially source of daidzein within PLRE. Therefore, the ratio *Q_DEP-extract_* was proposed and calculated by dividing the amount retained in the membrane by the sum of the amount of daidzein (**23**) measured in the donor and the membrane after 100 min of experiment (Section 4.10.2, Equation (9)).

In contrast to the untargeted experiments, for targeted quantitative measurements, the samples were collected every 20 min throughout the experiment, which was necessary to calculate the permeability coefficient (*P_app, pig_*) (Equation (1)). All samples were then analyzed by UHPLC-MS-TQ-MRM (Section 4.8). The following sections present the quantification results in detail.

#### 2.6.1. Permeation of Puerarin

The targeted quantitative measurement confirmed the semi-quantitative observations on puerarin. Indeed, puerarin crossed the membrane with an apparent permeability (*P_app, pig_*) (Equation (6)) of 2.61 ± 0.41 × 10^−6^ cm/s when present in the extract and with a higher *P_app, pig_* of 9.62 ± 1.56 × 10^−6^ cm/s when used alone. The ratio *Q_DEP_* (Equation (7)) provides information on the amount of puerarin retained in the membrane. They were 0.22% in PLRE and 0.47% for the pure standard, which is included in a range indicating the absence of accumulation in the membrane [29] (Appendix A).

#### 2.6.2. Permeation of Daidzein and Genistein

First, the permeation experiment performed with the *O*-glucoside standard daidzin (**8**) confirmed its cleavage into the aglycone daidzein by the membrane. As in the semi-quantitative experiment, neither daidzin nor daidzein was observed in the acceptor compartment, and daidzein was detected in the extracted membrane (Appendix A). The amount of daidzin remaining in the donor compartment after 100 min was measured at 40% of its initial amount. The ratio *Q_DEP_* (Equation (7)) between daidzein measured in the membrane and daidzin’s initial amount indicated that 0.63% was retained in the membrane. This slightly higher value than the *Q_DEP_* of puerarin can be explained by the higher lipophilicity of daidzein (Table 1).

The quantification results for PLRE demonstrated that the initial concentration of daidzein (138 ± 3 nM) increased eleven times to reach 1.544 ± 0.197 µM. This final concentration was used to calculate the ratio *Q_DEP-extract_* to evaluate the proportion of the aglycone daidzein in the membrane after 100 min of the experiment (Equation (9)). This ratio reached 5.38 ± 0.96% (Appendix A).

For the second aglycone, genistein (**28**), its trace amount detected in PLRE (below 5 nM (LOD)) increased to 121 ± 15 nM after 100 min of experiment. The ratio *Q_DEP-extract_* (Equation (9)) of genistein highlighted that 21.94 ± 2.69% of genistein was present in the membrane. This high ratio compared to that of daidzein could be related to its lower concentration range, as well as its higher lipophilicity (Table 1 and Appendix A).

## 3. Discussion

### 3.1. Comparison between Ex Vivo Permeation Results and Plasma Data in Humans

PL was selected because it contains various isoflavones that have already been investigated in different PK studies in humans. These isoflavones did not always originate from PL only but also other Fabaceae. Indeed, several studies have focused on soy isoflavones, including daidzin (**9**), daidzein (**23**), genistin (**16**), and genistein (**28**), because of the great interest in diets containing soya [47]. Furthermore, red clover (*Trifolium pretense* L.) contains other isoflavones, formononetin (**30**) and biochanin A (**32**), for which previous PK studies were performed [49,60]. Concerning puerarin (**5**) which is specific to the genus Pueraria [54], its PK parameters were studied following ingestion of PL extract [61,62]. These studies measured the classic PK parameters describing the plasma circulation, such as the maximal concentration and the terminal half-time of given isoflavones, which considered the bioavailability resulting from the various PK steps.

In the next sections, we discuss our results in comparison with PK data available in human studies for the different type of isoflavones.

#### 3.1.1. Fate of *C*-glycosides

The detection of *C*-glucoside puerarin (**5**) in the acceptor compartment and the determination of its *P_app, pig_* indicated that puerarin (**5**) is a highly permeable substance according to the Biopharmaceutical Classification System (BCS class I/II) [63]. Such data were consistent with the observation that puerarin (**5**) was present in human plasma at a biologically relevant level for pharmacological activity after oral administration of PL extract [61]. Another human PK study also monitored both levels of daidzein (**23**) and puerarin (**5**) following ingestion of PL and demonstrated the concomitant presence of both puerarin and daidzein in the plasma circulation [62]. The deglycosylation of puerarin to daidzein was assumed following in vivo studies in rats, which observed daidzein in the urine after ingestion of puerarin [45,64]. In our study, compared to the *O*-glycoside levels, the puerarin concentration in the donor compartment was not affected by the porcine membrane, indicating that no active enzyme for *C*-glycoside hydrolysis is present in the porcine jejunum, which is logical since *C*-glycosides are known to be hydrolyzed only by the microflora of the colon [65,66].

The other *C*-glycosides were also notably unaffected by the membrane in the donor compartment, and two of them were detected in the intestinal membrane (6″-xylopuerarin (**7**) and 3′-methoxypuerarin (**8**)) (Table 1 and Table 3, and Appendix A). Although further human PK studies are required, our observations indicated that their fate appears to be similar to puerarin, which is in line with a study in rats in vivo [67].

#### 3.1.2. Fate of *O*-glycosides

No permeation of the intact *O*-glucoside isoflavones was observed, and, in contrast, there were significant levels of their corresponding aglycones in the membranes. These results were consistent with previous reports of the cleavage of *O*-glucosides in the small intestine in studies with soya, which demonstrated that *O*-glycoside isoflavones could be cleaved in vitro using cell-free human small intestine extracts [68]. Hydrolysis of *O*-glycosides by gut microflora was also described [65].

Nevertheless, the human bioavailability of aglycones versus the “parent” *O*-glycosides has been a controversial issue [69]. For example, some human PK studies suggested that *O*-glycosides increased aglycone bioavailability after ingestion [49,50,52], while another suggested that differences in bioavailability were not very significant [51]. On the other hand, after deglycosylation and absorption, isoflavone aglycones are subjected to conjugation by phase 2 metabolism, as well as to an enterohepatic cycle. Thus, daidzein and genistein were observed in plasma mainly as conjugated metabolites [49,70].

In our study, daidzein and genistein were only detected in the membrane and not in the acceptor compartment. In comparison to previous studies describing their plasma circulation, we can assume that their passage through the intestine is ongoing in our experimental setting. Drug retention in the membrane is frequently observed and may be explained by the limited duration of the experiment, due to the need to ensure the intestinal membrane’s viability [24,27,29].

Other aglycones (formononetin (**30**), biochanin A (**32**), and 3′-methoxydaidzein (**24**)) resulting from the cleavage of *O*-glycosides (respectively, **22**–**25**, **27, 10**) were observed in the intestinal tissue (Table 1, Table 2 and Table 4). Formononetin (**30**) and biochanin A (**32**) are the 4′-methylated equivalents of daidzein (**23**) and genistein (**28**), respectively (Table 2 and Table 4). Their PK behaviors were investigated in studies related to red clover [49,60]. Their plasma levels were low due to their rapid demethylation into daidzein and genistein, which in turn reached relevant plasma concentrations [49,60]. Hepatic cytochromes, mainly CYP1A2, appear to be responsible for this demethylation [71]. Concerning 3′-methoxydaidzein (**24**), its plasma circulation was observed in an in vivo study in rats after ingestion of a TCM multi-herb formula containing PL [46]. However, 3′-methoxydaidzein (**24**) could undergo demethylation as formononetin (**30**) and biochanin A (**32**). In our experimental setting, no demethylation was observed so far, which indicates that no enzymes responsible for this biotransformation are present in the jejunum. Compared to in vivo data, it has to be kept in mind that the results described here are limited to what happens in the jejunum and probably underestimates the amount of aglycones since *C* and *O*-glycosides will be more exhaustively hydrolyzed by the intestinal microflora [65]. On the other hand, methylated isoflavones may be demethylated by hepatic metabolism [71]. Considering the chemical composition of PLRE, this in vivo biotransformation may further increase the amount of daidzein in the plasma. Finally, another aglycone (**31**) increased in the presence of the intestinal membrane. It has been annotated as tithonin, which is a di-methylated aglycone. Further studies are required to understand its origin.

In this study, the jejunum was employed as the most representative segment for absorption in the small intestine. However, the Ussing chamber system enables investigation of regional variation in intestinal absorption since duodenum, ileum or indeed colon can be studied [29]. It is commonly accepted that the small intestine is the principal absorption segment [72], but the colon is also a site of absorption depending on the substances studied, and further studies are necessary to assess its role [73,74].

### 3.2. Enzymatic Activities and Active Transporters

The experimental set up used here ensured the intestinal tissue’s viability and had the advantage of maintaining the activity of enzymes and transporters found in the gut wall [29,38]. Besides passive permeation, active transports, notably of *C*-glycoside compounds, and biotransformation of substrates were observed.

#### 3.2.1. Hydrolysis of *O*-glycoside Isoflavones

Our results indicated the presence of enzymes capable of hydrolyzing *O*-glycoside isoflavones in the jejunum. The enzymes responsible for this cleavage in humans are commonly referred to as β-glucosidases, which comprise Lactase Phlorizin Hydrolase (LPH), present in the apical membrane, and Cytosolic β-Glucosidase (CBG), located in the enterocytes [68]. The location of CBG implies that *O*-glycosides have to be transported to the cells before being hydrolyzed, while, for LBH, no transport is necessary. Previous studies have reported that either CBG [68] and/or LPH [75,76] are involved in isoflavone glycosides biotransformation. Since these studies are controversial, our ex vivo model could be used for studying this process in more detail.

Interestingly, and in contrast to the *O*-glycoside’s cleavage for the majority of compounds, an exception was observed with the malonyl-glycoside isoflavones (**17**, **20**), which remained much more stable than the other *O*-glycosides (Table 4). An in vivo study in rats comparing the plasma circulation of daidzein (**23**) from malonylglucoside-daidzin and daidzin had already observed the protective role of the malonyl group against enzymatic hydrolysis [77].

#### 3.2.2. Intestinal Absorption of Isoflavones

Our results indicated that *C*-glycosides and aglycones crossed the intestinal barrier. The permeation of puerarin is most probably an active transport since it was not found to be a passive permeant compound using the PAMPA model [13]. The transporters responsible for the entry of isoflavones into cells are controversial. A first hypothesis highlighted the role of the Sodium-dependent Glucose Transporter (SGLT1) [75], which was refuted in another study [78]. The potential involvement of the Glucose uniport (GLUT) family transporters have also been suggested without a clear answer so far [78]. SGLT1 is known to transport glucose actively in enterocytes. The glucose is then passively transported into the systemic circulation by the facilitated diffusion transporter GLUT2 [79]. Studies on living pigs have shown the presence of these transporters in all three segments of the small intestine [80]. Thus, our ex vivo model could be an attractive model to study isoflavone transport mechanisms in detail.

#### 3.2.3. Enzymatic Hydroxylation of Isoflavones

Another exciting observation noted in our study is the increase of 3′-hydroxypuerarin (**3**) and other tri-hydroxylated isoflavones (**4**, **18**, **24**) in the donor compartment (Table 1, Table 2 and Table 3). This observation could be explained by the presence of CYP3A4, which is responsible for phase 1 hydroxylation and was previously described as active in the porcine small intestine in the same setting [29]. Our observations also confirmed the viability of CYP3A4 when analyzing an HP, such as PLRE.

## 4. Material and Methods

### 4.1. Chemicals

Puerarin, daidzin genistin, daidzein, and genistein (all >98%) were purchased from Biopurify Phytochemicals Ltd. (Chengdu, China). Agar, calcium chloride dihydrate, glucose hydrate, magnesium chloride hexahydrate, potassium chloride, sodium chloride, sodium phosphate monobasic, and sodium hydrogencarbonate were obtained from Hänseler AG (Herisau, Switzerland).

### 4.2. Plant Material, Extraction and Sample Preparation

The dried root of *Pueraria montana* var. *lobata* (Willd.) Sanjappa & Pradeep (PLR) was acquired from Herba Sinica (Ch.B. 130301H083, Rednitzhembach, Germany) and tested for identity, purity, and residues and organic traces by Dr. Uwe Gasser, (Sebastian Kneipp research laboratory, Bad Wörishofen, Germany). The plant name was verified with www.theplantlist.org on 6 February 2019. The extract (PLRE) was prepared by aqueous extraction, which corresponds to the traditional decoction used in TCM [43]. The powdered herb was boiled 40 min in distilled deionized water at a ratio of 1 g for 10 mL. After cooling to 60 °C, the solution was filtered, and the residue was directly extracted a second time with the same volume of water. The two filtrates were combined and freeze-dried for 48 h (−80 °C, 0.01 bar, Christ Alpha 2–4 LD plus, Osterode am Harz, Germany), resulting in a yield of 50.2% (*w*/*w*) of PLRE. The freeze-dried extract was stored at −20 °C for the duration of the study.

For UHPLC-UV-PDA-HRMS/MS and UHPLC-UV-PDA-ELSD metabolite profiling, PLRE was solubilized in a solution of water and methanol (7/3 *v*/*v*) at a concentration of 5 mg/mL. The solution was sonicated (15 min) and centrifugated (10 min, 6000 rpm) (Prism R, Labnet International, Inc., Edison, NJ, USA).

For intestinal permeation experiments, PLRE was solubilized in Krebs-Bicarbonate Ringer buffer (KBR) (Section 4.3) at 200 µg/mL, and pure isoflavones at 100 µM. These solutions were prepared 1 h before the experiments, sonicated 15 min, and maintained at 38 °C.

### 4.3. Porcine Intestinal Tissue

This protocol was mainly based on and adapted from a previous study [29]. Jejunum from 6-month-old female Swiss noble pigs (weight 100–120 kg) was supplied by two local slaughterhouses (Abattoir de Meinier, Meinier Switzerland, Abattoir de Loëx, Bernex Switzerland). The tissue was collected directly after slaughter and rinsed with ice-cold KBR (120 mM NaCl, 5.5 mM KCl, 2.5 mM CaCl_2_, 1.2 mM MgCl_2_, 1.2 mM NaH_2_PO_4_, 20 mM NaHCO_3_, and 11 mM glucose; pH 7.4) [81] to remove debris from the jejunum. During transport from the slaughterhouse to the laboratory, the tissue was stored in the ice-cold KBR and constantly bubbled with a 95% O_2_-5% CO_2_ gas mixture (PanGas AG, Dagmersellen, Switzerland). In the laboratory, the jejunum was opened along the mesenteric border and rinsed with ice-cold KBR, according to a previously published protocol [82,83]. Scalpel and fine forceps were used to remove the muscle layer carefully. The remaining tunica mucosa and submucosa were cut into pieces of approximately 1.5 cm^2^.

### 4.4. Ussing Chamber Setup and Procedures for Intestinal Permeation Experiments

The permeation studies were performed on a six Ussing chamber system linked to a VCC MC6 MultiChannel Voltage-Current Clamp (Physiologic instrument, San Diego, CA, USA) with a heating block and six input modules with integral dummy membranes. The temperature was regulated by a circulating water bath (ED-5, Julabo GmbH, Seelbach, Germany). In accordance with [82], the Ussing chambers were prepared by putting first the Ag/AgCl electrodes in tips containing a mixture of 3% agar in 3 M KCl, which were then inserted into the Ussing chambers. Donor and acceptor compartments were filled with preheated KBR (38 °C, body temperature of pigs) that was constantly bubbled with the 95% O_2_-5% CO_2_ gas mixture. This gas mixture allowed the buffer to be mixed and circulated, in addition to oxygenating the intestinal tissue. Any voltage difference between the electrodes of the acceptor and donor compartment due to the buffer solution was eliminated. The chambers were then emptied, and the prepared jejunum was mounted on the sliders, which were inserted into the Ussing chambers. The exposed surface area of the jejunum was 1.26 cm^2^. Considering the travel time from the slaughterhouse and these preparation steps, the tissues were inserted in the Ussing chambers 45 min after the slaughter of the pigs. To minimize the potential interference of endogenous material, a 30 min equilibration period was performed by adding 7 mL of KBR to the donor and acceptor compartments to allow the endogenous substances to be released. In addition, to evaluate these potential interferences of endogenous substances, pieces of prepared jejunum were put into 7 mL of KBR in separate tubes and maintained at 38 °C for the duration of the experiment, to obtain blank membrane solutions.

After the equilibration period, both compartments of the Ussing chambers were emptied, and the acceptor compartment was filled with the same volume of fresh, preheated KBR. At the same time, the donor compartment was filled with preheated KBR solutions (7 mL) containing PLRE (200 µg/mL) or pure isoflavones (100 µM). These solutions were prepared 1 h before the experiments, and maintained at 38 °C.

During the experiment, the viability of the jejunum was monitored by measuring the TEER according to Ohm’s law. To this end, the voltage variation was recorded during the application of a 50 µA current pulse (duration 200 ms) each minute. Tissues with a TEER below 15 Ω cm^2^ were considered non-viable or non-intact and no analyses were performed on the samples collected from these Ussing chambers.

All experiments were performed in quadruplicate. In total, four experiments were conducted. With PLRE, two experiments were performed, one for untargeted UHPLC-HRMS metabolite profiling and the other for targeted UHPLC-MS-TQ-MRM quantitative analyses. Two experiments were carried out with pure substances (puerarin and daidzin) for targeted quantitative analyses. During the quantitative experiments, aliquots were collected from both compartments. In acceptor compartments, 400 µL were sampled and replaced by fresh KBR. In donor compartment, 200 µL were collected for PLRE and 100 µL for pure substances. The volume was replaced by the stock solution maintained at 38 °C. For quantitative analyses, aliquots were collected at 0, 10, 20, 40, 60, 80, and 100 min for PLRE, and an additional point at 120 min was collected for the pure substance. For untargeted metabolite profiling of PLRE, aliquots were collected at 0 and 100 min. The blank membrane solutions were also collected after 100 min for PLRE, and after 120 min for the pure substances.

After completion of the experiment, the jejunum slices employed in the Ussing chambers were collected, cut into small pieces, and extracted for 6 h using the mobile phase used for UHPLC analytical method (Section 4.4) in order to measure the potential amount of PLRE constituents or pure substance retained in the jejunum.

### 4.5. UHPLC Chromatographic Condition

The same chromatographic conditions were used for targeted, untargeted, and semi-quantitative analyses. For pure compound experiments, samples were injected (2 µL) into an Acquity UPLC^®^ BEH C18 column (1.7 µm, 2.1 × 50 mm; Waters, Milford MA, USA) and eluted (0.5 mL/min, 40 °C) with water (A) and acetonitrile (B) both with 0.1% formic acid (Fischer Chemical, Fischer Scientific AG, Reinach, Switzerland). A 2-min gradient of 10 to 98% of B was applied, followed by an isocratic step of 0.8 min at 98% of B and a re-equilibration step of 2 min.

For PLRE experiments, samples were injected (2 µL) into an Acquity UPLC^®^ BEH C18 column (1.7 µm, 2.1 × 100 mm; Waters, Milford, MA, USA) and eluted (0.5 mL/min, 40 °C) with water (A) and acetonitrile (B) both with 0.1% formic acid. The following gradient was used: from 10 to 17% of B from 0 to 5 min, 17 to 75% from 5 to 11 min, 75 to 98% from 11 to 12 min, an isocratic step at 98% for 2 min, and a re-equilibration step of 2 min.

### 4.6. UHPLC-UV-PDA-ELSD Semi-Quantitative Metabolite Profiling of PLRE

Semi-quantitative metabolite profiling of PLRE was obtained in a UHPLC system coupled to UV-PDA and ELSD. This system was controlled by Empower Software v2.0, its UV-PDA acquired from 200 to 500 nm (1.2 nm of resolution), and ELSD Sedex 85 (Sedere LT-ELSD, Alfortville, France) was set at 45 °C with a gain of 8. Chromatographic traces were exported from the proprietary format to text files to be plotted on Prism 8 (GraphPad Software, Inc., San Diego, CA, USA).

### 4.7. Untargeted UHPLC-HRMS/MS Metabolite Profiling of PLRE

This method was first used to analyze PLRE before the permeation experiment to evaluate its chemical composition. Five isoflavone standards previously described in PLRE [43] (puerarin (**5**), daidzin (**9**), genistin (**16**), daidzein (**23**), and genistein (**28**)) were analyzed simultaneously. The same method was then used for monitoring PLRE constituents in permeation experiments. In this monitoring, all samples (stock solution, donor and acceptor compartments, and extracted membrane) were initially acquired in UHPLC-HRMS mode only. Then, the stock solution and donor compartment were acquired by UHPLC-HRMS/MS. The details of the data acquisition are presented in the Appendix A. The obtained chromatograms were exported to Prism 8 software to be prepared for the figures. ThermoRAW MS data were converted to mzML using ProteoWizzard [84].

### 4.8. Data Processing of the Metabolite Profiling

A data processing strategy was designed to interpret the metabolite profiling to (i) assess the chemical composition of PLRE and (ii) consider the expected low concentrations in the acceptor compartments and the extracted membranes (see Appendix A for a scheme of the data processing workflow).

#### 4.8.1. Characterization of the Chemical Composition of PLRE

To assess the chemical composition of PLRE, *mzML* files of PLRE, solutions of isoflavone standards and adequate blank solutions were loaded into MZmine 2.38 [85,86]. MS features were extracted and deconvoluted into peaks following the “Automated Data Analysis Pipeline” (ADAP) workflow [87] with a tolerance range of 5 ppm, followed by a deisotoping step set at 80% and an alignment step (Appendix A for the precise peak-picking and data processing parameters).

A customized database was prepared from the entries of the species *Pueraria lobata* referenced in the Dictionary of Natural Products (DNP) [54] to annotate the resulting peak list at HRMS level, i.e., at the molecular formula level. The peak list was also annotated by adduct and complex search modules contained in MZmine. This annotated peak list was filtered to keep only peaks with HRMS/MS spectrum detected in PLRE. The HRMS data, containing the peak heights, retention time, and HRMS annotations, was exported to a text file, whereas the HRMS/MS spectra were exported as an .mgf file to be submitted to the online workflow at Global Natural Products Social Molecular Networking (GNPS) to generate a feature-based molecular network (FBMN) [41,42] (Figure 2B) (Appendix A).

Molecular network (MN) was created where edges were filtered to have a cosine score above 0.7 and more than 6 matched peaks. Further edges between two nodes were kept in the network if and only if each of the nodes appeared in each other’s respective top 10 most similar nodes. Tolerances for precursor and fragment ions were set at 0.02 Da. The spectra in the network were then searched against GNPS spectral libraries. All matches kept between network spectra and library spectra were required to have a cosine score above 0.7 and at least 6 matched peaks. The GNPS jobs parameters are available on https://gnps.ucsd.edu/ProteoSAFe/status.jsp?task=956dbd0a98cb4c0bb161248330173fd6 (accessed on 8 July 2021) for the FBMN presented in the Appendix A, and on https://gnps.ucsd.edu/ProteoSAFe/status.jsp?task=053e7abf891f4bef899a87abba66ab6d (accessed on 8 July 2021) for the FBMN in Appendix A.

The next step was to compare the experimental spectra organized in the MN against an in-house in silico Database (ISDB) of in silico fragmented spectra prepared from the DNP, according to a previously reported workflow [40]. A top 50 consultation was used for annotating the peaks at a tolerance of 0.005 Da and a spectral score threshold of 0.2. These top 50 annotations were then subjected to a taxonomic weighting script that complemented spectral similarity score with an additional score based on previously reported occurrences in the species, genus and family of PL in the DNP [55]. A top 6 annotation was kept after this step.

FBMN were visualized with Cytoscape 3.7.0, and annotations were displayed, thanks to the Chem Viz 2 1.1.0. Tables from HRMS data, GNPS library, and taxonomically weighted ISDB annotation were loaded in Cytoscape. Clusters that contained the isoflavones standards were further exported into Prism 8 (GraphPad Software, Inc., San Diego, CA, USA). Peak height intensity and retention time were used to plot the peaks included in selected clusters as a bar plot, named bar chromatogram (Figure 1D and Figure 2B–D). The prediction of the partition coefficient LogP [58] was used to complete the annotations, notably to discriminate between potential isomers.

Through this annotation process, 24 constituents of PLRE were putatively identified, including five isoflavones whose standards had been analyzed (Table 1). It should be noted that genistein was present in trace amounts in PLRE.

#### 4.8.2. Data Processing of Metabolite Profiling from Permeation Experiments

For the metabolite profiling obtained with aliquots collected during the permeation experiment, a low concentration was expected: first, the stock solution of PLRE was prepared at 200 µg/mL and second, the concentration differential between the donor and acceptor compartment is large. To best consider these low concentrations, detection limits were evaluated. Thus, the limits of detection (LOD) and quantification (LOQ) were measured for the five standards analyzed by UHPLC-HRMS only and used to optimize the parameters for peak detection (Appendix A). For this purpose, the peaks corresponding to the five standards were extracted and deconvoluted thanks to the algorithm “Targeted peak deconvolution” integrated into MZmine, which selectively checks peak (*m*/*z* and retention time) according to a customized list of potential peaks. After targeted peak picking, the peak height intensities were used to measure the LOD and LOQ of puerarin, daidzin, genistin, daidzein, and genistein. LOD and LOQ were calculated using the standard deviation of the response and the slope method according to the international conference of harmonization (ICH Q2(R1)). The standard deviation of y-intercepts of regression lines was used as the standard deviation and was calculated with the LINEST function in Microsoft Excel 16.16.8. The LOD-LOQ were measured on the day of the analyses of the PLRE experiment with concentrations from 2 to 50 nM acquired in triplicate and were determined as follows: 2–5 nM for puerarin, 2–7 nM for daidzin, 5–16 nM for genistin, 2–5 nM for daidzein, and 3–8 nM for genistein.

These concentration ranges corresponded to a minimum peak height of around 1.5 × 10^4^ and a number of acquisition point of more than 20 spectra. These values were used for an untargeted peak picking with the ADAP workflow of all metabolite profiling from the permeation experiments (Appendix A). However, the peak height threshold needed to be increased to 5 × 10^4^ because the computational resources necessary for the deconvolution of the peak height values defined by the LOD were too demanding. After a peak picking considering this limitation, the peak lists acquired with HRMS only were subsequently aligned and exported for interpretation. A targeted peak picking was also performed for the 24 annotated constituents of PLRE to overcome computational limitations and improve the detection capacities of the peak picking. (Appendix A).

A first interpretation of the constituents’ fate in the acceptor compartment and the extracted membranes indicated that biotransformation processes occurred in the PLRE metabolome once in contact with the jejunum (Section 2.5.1). Therefore, a second set of annotations was performed with the most concentrated aliquots of the permeation experiment. The stock solution, the donor compartment, and the adequate blank solutions acquired by UHPLC-HRMS/MS, were peak-picked by adapting the parameters defined for HRMS only (Appendix A). These metabolite profiles were used to generate a second FBMN, as described previously. This FBMN provided a very efficient visual support to interpret the changes occurring in the PLRE metabolome in the donor compartment once in contact with the porcine intestinal membrane (Appendix A). This second FBMN intended to annotate substances that appeared or increased significantly in the donor compartment. This second round of annotations permitted to identify putatively eight additional NPs (Table 1).

The final steps in data processing consisted of establishing a peak list of all annotated NPs representing PLRE constituents and their intestinal metabolites. This peak list was used in a final peak detection round with the “Targeted peak algorithm” (Appendix A).

For both peak picking methods, the fate of each peak was estimated in a semi-quantitative manner by calculating the ratio between the HRMS peak height in the different compartments and expressed as a percentage (Equations (1)–(5)) (see Appendix A for the calculated ratios in each compartment).
(1)$$\mathrm{Rati}o_{ACC_{100min}/T_{DON\;0\;min}}=\frac{Peak\;height_{\mathrm{accepteur}\;100min}}{Peak\;height_{\mathrm{donor}\;0\;min}}\;\cdot 100\;\left[\%\right],$$
(2)$$\mathrm{Rati}o_{ACC_{100min}/T_{DON100\;min}}=\frac{Peak\;height_{\mathrm{accepteur}\;100min}}{Peak\;height_{\mathrm{donor}\;100\;min}}\;\cdot 100\;\left[\%\right],$$
(3)$$\mathrm{Rati}o_{M_{100min}/T_{DON0\;min}}=\frac{Peak\;height_{\mathrm{extracted}\;\mathrm{membrane}\;100min}}{Peak\;height_{\mathrm{donor}\;0\;min}}\;\cdot 100\;\left[\%\right],$$
(4)$$\mathrm{Rati}o_{M_{100min}/T_{DON100\;min}}=\frac{Peak\;height_{\mathrm{extracted}\;\mathrm{membrane}\;100min}}{Peak\;height_{\mathrm{donor}\;100\;min}}\;\cdot 100\;\left[\%\right],$$
(5)$$\mathrm{Rati}o_{DON_{100min}/T_{DON0\;min}}=\frac{Peak\;height_{\mathrm{donor}\;100min}}{Peak\;height_{\mathrm{donor}\;0\;min}}\;\cdot 100\;\left[\%\right].$$

Eventually, peak height intensity and retention time of clusters of interest in the FBMN were exported to plot the clustered peaks as a bar chromatogram with Prism 8 (GraphPad Software, Inc., San Diego, CA, USA) (Figure 1D and Figure 2B–D).

### 4.9. UHPLC-MS-TQ-MRM Quantitative Analyses

Quantitative measurements were implemented to calculate the permeability coefficient of the absorbed constituents, as well as accurately monitor the amount of some constituents in PLRE. The quantitative measurements were performed with a UHPLC system connected with a triple quadrupole (TQ) mass spectrometer operating in multiple reaction monitoring (MRM). The details of the data acquisition are presented in the Appendix A.

The limit of detection (LOD) and quantification (LOQ) were measured with the same method than in UHPLC-HRMS, at the difference that the peak areas were used instead of the peak heights. The LOD-LOQ were measured the day of the PLRE experiment with concentrations from 5 to 50 nM and were determined as follows: 3–10 nM for puerarin, 3–8 nM for daidzin, 3–8 nM for genistin, 20–60 nM for daidzein, and 12–37 nM for genistein.

### 4.10. Permeation Data Analysis

#### 4.10.1. Apparent Permeability Coefficient

The apparent permeability coefficient (*P_app, pig_*) relates the rate at which a given substance crosses the membrane [14]. It was calculated as follows:(6)$$P_{app,pig}=\frac{dc}{dt}\cdot \frac{V}{A\cdot C_{0}}\left[\frac{cm}{s}\right],$$
where *dc/dt* is the slope of the concentration-time curve between 20 and 80 min, *V* is the volume in the donor compartment (7 mL), *A* is the exposed surface area (1.26 cm^2^), and *C*_0_ the initial concentration in the donor compartment (µM).

#### 4.10.2. Evaluation of the Retention of NPs in the Intestinal Membrane

The concept of the Transport index (*TI*) was recently proposed to consider the retention of compounds in the intestinal membrane [24]. *TI* is defined as the sum of the quantity of substance transported in the acceptor compartment (permeating quantity *Q**_PERM_*) and accumulated in the intestinal tissue (quantity deposited *Q**_DEP_*). *Q**_PERM_* and *Q**_DEP_* are expressed as a percentage of the amount deposited in the donor compartment at the beginning of the experiment. In this study, their calculation was adapted in two ways. First, for the pure substance, the calculation was performed in molar quantity to compare NPs in molar relationships:(7)$$Q_{DEP}=\frac{nmole_{membrane\;100min}}{nmole_{donor\;0\;min}}\;\cdot 100\;\left[\%\right],$$
(8)$$Q_{Perm}=\frac{nmole_{acceptor\;100\;min}}{nmole_{donor\;0\;min}}\;\cdot 100\;\left[\%\right].$$

Second, for PLRE, the calculation aimed at considering the biotransformation observed in the donor compartment after 100 min of experiments. Indeed, some intestinal metabolites retained in the membrane could come from several constituents of PLRE. In order to estimate the quantity deposited in the membrane in a relevant way, we propose to evaluate this quantity in relation to the sum of quantity measured in the donor at 100 min and in the membrane at the end of the experiment by defining a ratio of quantity deposited specific to the extract (*Q**_DEP-_**_extract_*):(9)$$Q_{DEP-extract}=\frac{nmole_{membrane\;100min}}{nmole_{donor\;100\;min}+nmole_{membrane\;100\;min}}\cdot 100\;\left[\%\right].$$

#### 4.10.3. Evaluation of the Biotransformation of NPs in the Intestinal Membrane

The metabolite formation index (MFI) was recently proposed to estimate the formation of metabolites in the presence of the intestinal membrane [59]. This calculation was only used to evaluate the formation of daidzein from the daidzin standard:(10)$$MFI=\frac{nmole_{donor\;120\;min}^{metabolite}+nmole_{membrane\;120\;min}^{metabolite}+nmole_{acceptor\;120\;min}^{metabolite}}{nmole_{Donor\;0\;min}^{precursor}}\cdot 100\;\left[\%\right].$$

### 4.11. Statistical Analyses

The data were expressed as the mean ± standard deviation (SD). The coefficient of variation (CV) was calculated and expressed as a percentage.

## 5. Conclusions

Porcine small intestine ex vivo was used to study the absorption and intestinal biotransformation of PLRE, selected for its diverse isoflavone content and the existence of previous metabolism studies on its constituents. First, a qualitative untargeted approach was attempted to observe the fate of as many constituents as possible. Observations were then verified and completed by a quantitative study.

This work demonstrated the ability of the porcine small intestine ex vivo to reproduce absorption and biotransformation processes with the PLRE. The untargeted approach monitored the fate of several PLRE constituents and identified a selection of compounds capable of crossing the intestinal barrier. The observations are consistent with previous reports on the absorption and intestinal metabolism of isoflavones, particularly with human studies. In the context of this study, it is clear that all of the NPs monitored were isoflavones. This was a consequence of the composition of the PLRE and the fact that we selected its decoction for this investigation. However, the methodology described here can be applied to any herbal decoction and should enable the monitoring of a broad range of NPs.

The combination of ex vivo porcine small intestine in the Ussing chamber system with untargeted metabolite profiling appears to be an interesting tool for highlighting the constituents of a plant extract that may reach the plasma compartment. Based on our observations, the intestinal membrane is fully viable, and active transport mechanisms and biotransformation can be observed in addition to passive permeation. Compared to PAMPA and Caco-2 cell line, this model has a lower throughput but, combined with generic untargeted metabolite profiling, provides precious data for estimating the absorption in NPs in complex extracts.

Thus, the approach identified NPs and their metabolites potentially capable of reaching pharmacological targets in vivo. Such information should be considered for conducting meaningful bioassays to explain the mode of action of the constituents of ingested HP. Compared to in vivo data in rodents, ethically, this model has the advantage of using largely available porcine tissue of pigs, slaughtered for nutritional reasons, and, thus, does not require laboratory animals’ sacrifice. Furthermore, the pig is a large omnivorous mammal and porcine intestine has greater physiological/functional similarity to the human intestine [29]. In further studies, we aim to apply our approach to various HPs with constituents of different chemical types to understand better their fate and use the data to conduct targeted pharmacological assays to decipher the mode(s) of action of such preparations.

## Figures and Tables

**Figure 2 metabolites-11-00541-f002:**
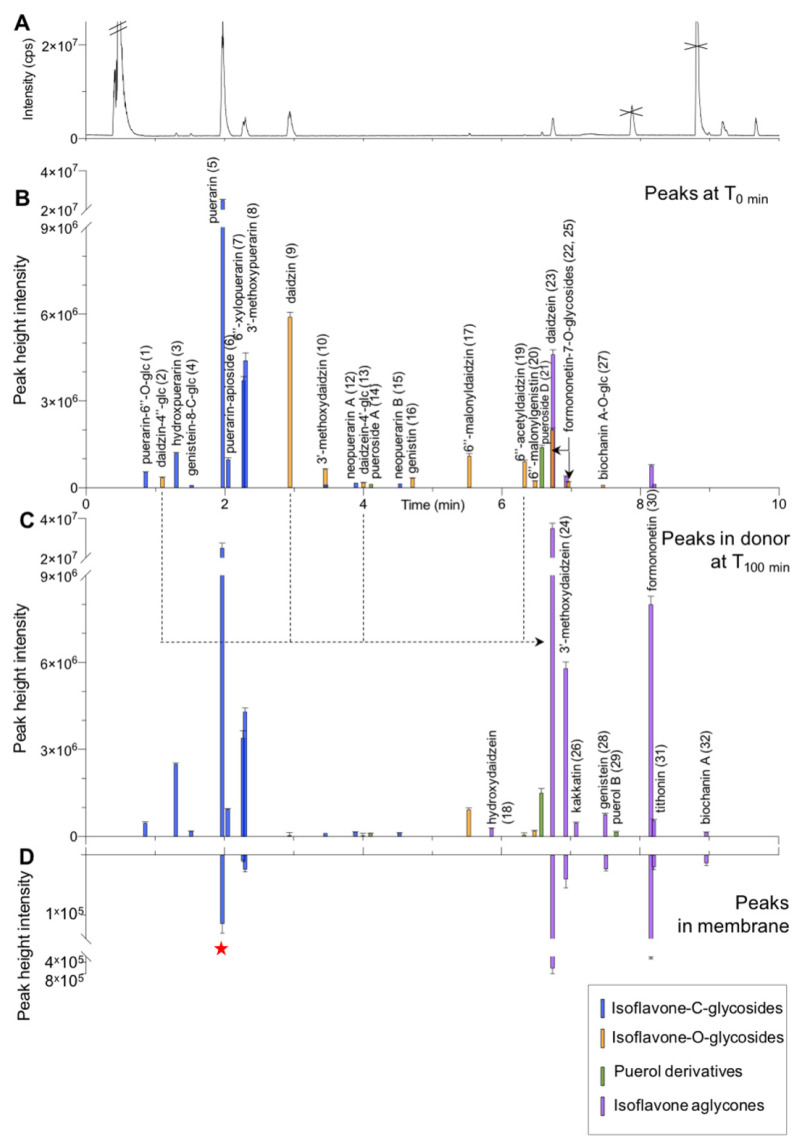
Monitoring of PLRE composition by metabolite profiling and peak picking during permeation experiments: (**A**) UHPLC-HRMS chromatogram of PLRE stock solution at 200 µg/mL. (**B**–**D**) Bar chromatograms of the annotated peaks, (**B**) in the stock solution (**C**) in the donor compartments at 100 min, and (**D**) peaks detected in the membrane. Peak heights are represented as mean ± SD, stock solution: *n* = 3 (analytical replicates), donor compartments and membrane *n* = 4 (experimental replicates). See Table 1, Table 2, Table 3 and Table 4 for the annotations. The red star on (**5**) indicates that this peak was also detected in the acceptor compartment.

## Data Availability

The dataset generated and analyzed for this study is available on the GNPS server (MassIVE MSV000087772 [doi:10.25345/C54N9Q] ftp://massive.ucsd.edu/MSV000087772/ accessed on 30 July 2021. It includes all raw UHPLC-HRMS data.

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
