# Peer review of "Using Porcine Jejunum Ex Vivo to Study Absorption and Biotransformation of Natural Products in Plant Extracts: Pueraria lobata as a Case Study"

_metabolites, 2021, doi:10.3390/metabo11080541_

Round 1
Reviewer 1 Report
In this study, the authors evaluated a positive intestine exvivo model in order to determine the metabolic fate of natural compounds, using Puraria lobata as case-study. The rationale for the study was properly supported, there was a good analytical approach and a careful discussion of the results.
However, my main concern is related to the number of identified compounds by an untargeted approach. The authors performed a quite complex data processing, which finally provided 32 isoflavones. Unless the authors decided to focus their study on isoflavones (which I did not clearly see in the manuscript), I am surprised that they did not identify other compounds, due to the high diversity of chemical entities that may be found in a natural product. Besides, although isoflavones are clearly the dominant phenolic compounds in Puraria lobata, for instance in soybean some phenolic acids and lignans have also been identified, so I would expect to find some of these compounds in an untargeted analysis. I wonder whether this is related to the fact that the authors chose to perform the analysis in positive mode, when it is known that phenolic structures are better ionized in negative mode.
Besides this general consideration, I have some particular comments:
- In Table 1, the authors should include the error they had in the identification of the compounds
- I find Tables 2-4 very useful for a better understanding of the results, but they are not mentioned in the text. The authors should correct this. Moreover, I think it would be better if current Table 4 would appear before the current Table 2 and Table 3.
- Page 3, lines 118-28. This may be deleted, since these methodological aspects are known by many readers and there is no a real need to include this explanation in the introduction, which is already rather long.
- Figure 1 is more a graphical abstract that does not need to be included in the manuscript, since the analytical strategy is clearly described.
- Page 16, lines 543-46. This is not completely right. Since about 15 years ago, it is known that colon is a key organ in the metabolic fate of polyphenols and, even for the most absorbed classes in the small intestine, as it is the case of isoflavones, about 50% of the original amounts reaches the colon.
Author Response
Please see the attachment (file named 2107230_JH_Metabolites_Absorption_R1_Reviewer1.pdf)

Reviewer 2 Report
The manuscript by Houriet et al, is reporting interesting results related to the intestinal absorption and biotransformation of NPs.
The major significance could be the porcine jejunum-Ussing system can resemble (or applicable to) human intestinal absorption (or transport) system, Unlike authors’ claim, genistein, daidzein, and others, could not cross the model membrane, as evident from zero concentration in donor cell. On the contrary, these are readily absorbed to human blood (one of many reports https://doi.org/10.1093/ajcn/77.2.411). So, there is a concern that the Ussing system used in this study can be applicable for the proposed study as authors claimed.
Others; please include more reports using Ussing system with other animal tissues, such as rat, dog and others, in the Introduction part.
Author Response
Please see the attachment (file named 2107230_JH_Metabolites_Absorption_R1_Reviewer2)

Reviewer 3 Report
In this manuscript it was presented the use of porcine jejunum ex vivo in Ussing chambers to evaluate the absorption and biotransformation of the plant extract (Pueraria lobata). In general, this manuscript it was excessively long. In the introduction there are several explanations that are not necessary (e. g. it seems that the objectives are repeated lines 129-134 and 150-154). Thus, maybe the lines 129-134 should be removed. In the following section (2. results) it seems that figure 1 fits better as a graphical abstract as seems to describe the main strategy of this work and is partially repeated with figure S4. Along the manuscript maybe other aspects could be transferred to supplementary data, etc. The references list also could be slightly reduced.
Along the manuscript sometimes it was difficult to understand all the ideas (e.g. in section 5.7.2 the authors presented 5 equations of ratios, but when I observe the supplementary tables the identification it was not clear and I did not find the values of the ratios as only seems to be presented three; in another example, in section 5.9.3 the authors explain the calculation of MFI in which was selected the 120 min but in the previous section it was referred that the study have 100 min and in the results section I did not can find the MFI value obtained!
Thus, globally I suggest the authors to carefully revise the manuscript in order to become the ideas more clear and more concise!
Author Response
Please see the attachment (file named 2107230_JH_Metabolites_Absorption_R1_Reviewers3.pdf)

Round 2
Reviewer 1 Report
I think the manuscipt has improved with the modifications included by the authors and it is now acceptable for publication.
Reviewer 3 Report
The manuscript should be accepted for publication.